# Developing a Model of Increasing the Learners' Bilingual Professional Capacity in the Virtual Laboratory Environment

**Aleksandra I. Dashkina, Ludmila P. Khalyapina, Aleksandra M. Kobicheva \*,**
**Maria A. Odinokaya and Dmitri A. Tarkhov**

Department of Advertising and PR, Peter the Great Saint-Petersburg Polytechnic University,
195251 Saint-Petersburg, Russia; dashkina_ai@spbstu.ru (A.I.D.); khalyapina_lp@spbstu.ru (L.P.K.);
odinokaya_ma@spbstu.ru (M.A.O.); tarhov_da@spbstu.ru (D.A.T.)
**\*** Correspondence: kobicheva_am@spbstu.ru

**Abstract:** The article considers industrial applications of digital twins and their contribution to decision-making and prevention of failures in manufacturing. Virtual laboratories are described as an example of using digital twins not only in industry but also for educational purposes. The article is also focused on the value of physical and virtual investigations through the prism of content and language integrated learning approach (CLIL) and it offers recommendations for combining the two to strengthen science inquiry learning and job-relevant learning. Students conduct online investigations resulting in interdisciplinary team interaction and knowledge integration. Adopting the CLIL approach changes engineering students' attitudes to learning a foreign language, drums up their academic interest, and strengthens their motivation. In addition, the article describes how neural networks are applied for the simulation of physical processes in a virtual laboratory. The advantages of using virtual laboratories for professional-oriented training include linking observable and unobservable phenomena, pointing out essential information, and providing online, adaptive guidance. To evaluate the efficiency of the virtual laboratory environment, we conducted an experiment to assess the effectiveness of an integrated learning course with the elements of a virtual laboratory. The sample was 48 students (23 of them were enrolled on the course based on integrated learning with elements of virtual environment; 25 students studied professional discipline and professional English separately). To collect the data two online testing on English were conducted. The professional discipline was evaluated once at the end of the semester. In addition, we conducted an online survey on motivation for students of both groups to reveal the influence of educational process organization on this indicator. The results of the experiment allowed us to ascertain that the CLIL approach in the virtual laboratory environment resulted in better outcomes and helped the students adopt a more positive attitude to learning English.

**Keywords:** CLIL approach; virtual laboratory environment; higher education; neural networks; simulation; integrated learning

---

## 1. Introduction

At present, polytechnic education takes into consideration recent scientific breakthroughs and disruptive innovations, including the ones related to Industry 4.0. Engineering students should keep up with the recent innovations so that they can draw on the knowledge they acquire at university in their future careers. To this end, they need to have a high level of foreign language proficiency because in their career they will take part in scientific conferences, work abroad, participate in international

projects, and read scientific articles. In this research, we intend to illustrate how students working in a virtual laboratory based on the digital twin technology in combination with the content and language integrated learning approach (CLIL) approach can gain knowledge in the subjects related to their profession by getting explanations, instructions, and assignments in English.

We will start our research by producing insights into various applications of digital twins in industry, in virtual laboratories in general, and in educational virtual laboratories in particular. We will also explore the advantages of a digital laboratory over a conventional one. Then we will go on to describe how the neural-network-based approach can be used for the simulation of physical processes in a virtual laboratory. In the second part of our research, we will consider the advantages of the CLIL approach and the ways in which it can be applied in a virtual laboratory. Finally, we will describe the experiment aimed at evaluating the efficiency of a virtual laboratory environment.

*1.1. Theoretical Background*

1.1.1. Digital Twins and Their Industrial Applications

Since the article mainly considers virtual laboratories, it is essential to produce some brief insights into digital twins because direct correlation can be established between the concept of a visual laboratory and the concept of a digital twin [1]. A digital twin is a virtual equivalent (prototype) of a real-world object, set of objects or processes; it is a computer model that duplicates the key characteristics of a physical object and that is capable of replicating its states under different conditions. It is a complex software product developed on the basis of various data. Its functions are not limited to collecting data at the stages of developing and making a product; it also collects and analyses data during the product lifecycle, sometimes using numerous Internet of things (IoT)-sensors. Digital twins emulate the operation of real-world devices. They involve developing virtual environments that allow testing different scenarios and getting the same results, which is faster, cheaper, and, to some extent, more reliable.

Basically, it is a number of mathematical formulae that describe the object itself as well as the processes running in it. The digital twin is based on developing and applying a number of complex multidisciplinary mathematical models described by 3D non-stationary nonlinear differential equations in partial derivatives. Such a twin can be developed even earlier than the physical object it is supposed to duplicate: a virtual model can be developed while the object (a building, an automobile, a device) is being designed. The technology allows simulating various situations that can arise in the workplace. Thus, the digital twin makes it possible to choose the best possible scenarios of technological processes so that failures and emergencies can be avoided. This research introduces the theoretical framework of the digital twin-driven assembly-commissioning method for high precision products with multidisciplinary coupling. A case study product was used to verify the effectiveness and feasibility of the proposed method.

In our research, a digital twin is regarded as a component in a complex system that represents the whole complex of many parts or even numerous systems and which can be presented as a range of digital technologies. These technologies apply approaches based on statistical analysis, machine learning, physics, control theory, reliability engineering, the theory of waiting lines, and numerical simulation of physical processes in the materials of an object. The latter allows simulating the behavior of complex systems by dividing them into numerous components, which are small enough for their properties to be considered as homogeneous ones. It is widely used for solving problems in such areas as deformable solid mechanics, thermal exchange, hydrodynamics, electrodynamics as well as streamlining the work.

Besides, CAD-models (computer-aided design/drafting and computer design aids) are widely used. They carry the information about the appearance and structure of objects, as well as the information about materials, processes, dimensions, and other parameters. FMEA (failure mode and effects analysis) models based on the analysis of system reliability are also applied. They are

capable of incorporating mathematical failure modes with the statistical database of failure modes. In fact, this methodology is used for performing analysis and determining the most critical phases of manufacturing processes.

Some experts divide digital twins into three types. The first one is referred to as the DTP (digital twin prototype), which is a virtual replica of a physical object available in the real world. The object contains the data required for the comprehensive description of the model, including the information about the way it was designed in real-world conditions. The second type called DTI (digital twin instance) gives the data needed for the description of a physical object, which usually includes an annotated 3-D model as well as the materials used currently and in the past, information about the components, the processes that take place in any period of time, test results, records about maintenance and repairs, operating data received from the sensors and monitoring parameters. The third type, referred to as DTA (digital twin aggregate) is a computing system that unifies all digital twins and their real-world prototypes and allows collecting and exchanging data. A digital twin allows using the minimum key parameters to reproduce all the other characteristics of an object. This technology can be used to solve different types of problems involving diagnostics of an object's state, forecasting, streamlining the work and control. Studies by Fei Tao et al., 2018, Grieves M. et al., 2017, Schmetz, A. et al., 2020 [2,3] reveal the essence of the digital twin concept, an emerging and fast-growing technology which connects the physical and virtual world. It should be noted that a multitude of terms exists describing similar or partially overlapping phenomena. For example, the term "digital shadow" is often used interchangeably with the digital twin—despite it is mainly referring to a digital footprint [4], digital counterpart [5], virtual twin [6], virtual object [7], product agent [8], and avatar [9]. The premise driving the digital twin concept was that each system consisted of two systems, the physical system that has always existed and a virtual system that contained all of the information about the physical system. Through modeling and simulation in a virtual laboratory, we can realize the emergent form and behaviors of systems and diminish the "I didn't see that coming" factor.

Digital twins can be used in various spheres. They can optimize performance by providing forecasts for long-term planning. In healthcare, they can capture and visualize hospital systems and provide doctors with digital visualization of different parts of the body. They are also capable of improving customer experience by creating a digital twin of the customer-facing applications, getting customer feedback, and improving services accordingly. They are able to improve maintenance by collecting performance data over time, analyzing them, and identifying the required maintenance. In machine building, while a machine is developed, its digital copy is created, which enables simulation and testing of ideas before actual manufacturing starts. Digital twins of real cities can be used to optimize urban sustainability [10]. Thus, digital twins will penetrate every nook and cranny of both manufacturing and other spheres of our life. They will dramatically change industries by boosting efficiency and preventing emergency situations.

The concept of a digital twin was introduced in 2003, as part of a university course on product lifecycle management; it was initially published in the aerospace field and was defined as "a reengineering of structural life prediction and management", but now more and more studies are conducted into possible applications of digital twins in various spheres. In the manufacturing sector, the digital twin can be regarded as a source of useful information across the entire product lifecycle, for instance, it applies to the building and infrastructure lifecycles. Therefore, the construction digital twin will be conducive to shifting from a smart and lean construction process towards smart lifecycle management [11]. By applying digital twins in their work, engineers can foresee and avoid possible problems as well as take cradle-to-grave responsibility for the constructions they design, taking into account both external and internal factors.

The digital twin can be endowed with an abstract model that has all characteristics and fully describes the physical twin at a conceptual level throughout the whole product life-cycle. This abstract model allows capturing and understanding as well as clearly describing the physical twin, its behavior,

and its environment at an abstract level [12]. Thus, digital twins will enable engineers, designers, and other stakeholders to assess the consequences of the decisions they make concerning products, processes, and services, as well as to improve the quality and performance of the product.

Cyber-physical production systems have considerable advantages over traditional ones and can be conducive to arriving at smart manufacturing solutions. In traditional production systems, physical manufacturing resources are registered in advance by manually establishing device dependencies, which is quite time-consuming and can result in errors. If unexpected changes take place in cyber-physical production systems based on digital twins, machines and sensors are dynamically registered, quick access to them is provided, and machines and sensors can be easily moved, added, or removed, which significantly improves monitoring and control of manufacturing operations [13]. We can see that the digital-twin approach will make manufacturing proactive rather than reactive, so the preventative maintenance of equipment will be more commonplace than dealing with failures. Digital twins and other technologies related to Industry 4.0 can improve all kinds of manufacturing practices. The integration of digital twin concepts into sustainable manufacturing can be regarded as a practical demonstration of how cutting-edge technologies contribute to the development of a circular economy, which can be defined as an economic system aimed at continual use of resources via recycling, reuse, sharing, etc. Using digital twins allows identifying possible machine states, identifying whether the machines need maintenance, measuring the consumption of power, and many other ways of improving efficiency when exploiting resources [14]. Thus, digital twins can be conducive to making environmentally friendly decisions, which is especially topical given the global environmental issues, such as pollution, waste management, and running out of non-renewable resources.

Thus, digital twins are widely used for monitoring and prediction as well as for helping humans in their decision-making. Some new technologies have been launched aimed at improving human-machine interactions, but humans have not yet become part of the smart manufacturing system. However, research is being done into creating digital twins for people in order to enable them to become part of the smart manufacturing environment [15]. Digital twins will bring about a paradigm shift in manufacturing systems by bridging the gap between the virtual and physical worlds. Virtual laboratories are one of the ways in which digital twins can be applied.

### 1.1.2. Virtual Laboratories vs. Conventional Laboratories

Virtual systems can simplify learning by highlighting salient information and removing confusing details such as obtaining the predicted desirable, eliminating the predictable undesirable, and decreasing the unpredicted undesirable [16], exploring unobservable phenomena, linking observable and unobservable phenomena, pointing out salient information, providing online, adaptive scaffolding guidance. Learners can conduct experiments about unobservable phenomena, such as chemical reactions, thermodynamics, or electricity. This enables learners to perform more experiments and thus to gather more information in the same amount of time it would take to do the physical experiment and to enable learners to conduct multiple experiments in a short amount of time. Moreover, learners realize how to extract valid information from a complex visualization in virtual experiments.

Virtual labs can be defined as an interactive environment where learners can draw on authentic experiences by conducting experiments. Students can access virtual laboratories to make direct observations, especially when getting access to a real laboratory is too expensive or even dangerous, like the Radioactivity iLab, in which students can measure radiation from a sample of radioactive material without being exposed to radiation [17]. Apart from creating a safer and cheaper environment, digital laboratories can help visualize the subject matter, which will improve learners' understanding. An experiment was conducted at senior high school in Saudi Arabia. It was aimed at measuring the impact of virtual labs on the educational science process. Working in the virtual laboratory really improved the quality of delivering information to the students, who were able to conduct the experiments individually. Moreover, the learners received adequate feedback during the learning process, and it was easier for the teachers to assess the students' performance [18]. If students

get accustomed to conducting experiments at school, the transition to the university level will be significantly easier for them because they will associate the laboratory-based work with a pleasurable experience. Anecdotal evidence suggests that not all students at university are willing to conduct laboratory experiments. Moreover, if they skip a class dedicated to laboratory-based work, they often have difficulty getting access to the laboratory in order to conduct the experiment individually. However, a virtual laboratory solves this problem because students can complete their laboratory assignments at their own pace.

In addition to applying digital twins in laboratory experiments, engineering students can be involved in developing a simulation model, its verification, and testing. In a study conducted at Helsinki Metropolia University of Applied Sciences groups of students participated in designing a 3-D model of mechatronics equipment and converting it into a digital twin. According to the results of the survey conducted after the experiment, the students who had participated in the project pointed out that their motivation for studying and their own responsibility for learning had increased. They also claimed that digital twin technology had made learning easier [19]. We believe that only undergraduate or even postgraduate engineering students can be involved in such major projects. Freshmen and sophomores are more likely to be users of digital twins rather than developers. For them working in a virtual laboratory can serve as a springboard for further research into digital twin technology and even developing digital twins of their own.

Consequently, freshmen and sophomores working in virtual laboratories improve their skills in recording, reporting, and interpreting observations; higher-level cognitive skills of deductive reasoning, hypothesis formation and testing, and skills related to manipulative and instrument use. A virtual laboratory can be used as an effective preparatory tool for gaining familiarity with the laboratory environment. It is especially vital for distance students, who, unlike on-campus students, have to complete laboratory assignments within a short period of time. However, some students indicate that a number of other problems associated with their laboratory sessions emerge during the study, including problems with mathematical calculations and ability to apply the theoretical concepts within practical tasks in the laboratory [20]. The results of the study conducted by Dalgarno, B., Bishop, A.G., Adlong, W., Beddgood Jr, D.R. indicate that in addition to their research based on hand-on experience in a virtual laboratory, they should seek to improve their knowledge of the familiarize themselves with the theoretical basic subjects, such as physics and mathematics. Research has been done into the ways of carrying out a simulation of various real-life processes in a virtual laboratory aimed at giving evidence that the neural-network-based approach has considerable advantages over other approaches.

### 1.1.3. Using Neural Networks for the Simulation of Physical Processes in a Virtual Laboratory

It is quite difficult to work remotely with a real (non-virtual) laboratory facility when several groups of students are supposed to do the assignment simultaneously because the connection can be bad and expensive equipment can be damaged. Therefore, it makes more sense to organize students' work with a local digital twin of a laboratory facility. This kind of digital twin requires a high-quality simulation of physical and any other processes in the facility.

Today simulation of physical processes is usually based on solving differential equations with boundary conditions. Generally, it is impossible to formulate exact solutions to such problems, so approximate solutions serve as a model. Differential equations, boundary conditions, etc. approximately conform with the simulated object, process, or phenomenon. A great number of conventional, proven approaches are available that make it possible to solve such problems. They include various versions of Gelerkin's method, finite-difference method, finite element method, boundary integral equations, asymptotic expansions, etc. [21–25]. Most of them are various types of numerical methods that allow obtaining pointwise or local solution approximation. Formulating an analytical form on the basis of a pointwise solution is a separate problem.

In conventional methods, a mathematical model in the form of differential equations, boundary conditions, etc. is regarded as a source object. If such a model describes the real object quite accurately

and thoroughly, such idealization is justified, but it does not hold true for the majority of technical applications, therefore, apart from them, observations are made. The latter should be taken into consideration in the course of the simulation. In this situation, the application for neural network modeling is justified [26]. The neural-network-based approach [27–36] makes it possible to obtain the solution immediately in the form of a function meeting the required smoothness conditions. It is even more important that neural networks are not affected by data errors and that natural parallelization of computations is possible.

It is necessary to mention a certain type of problems which require to develop a model for the whole interval at which the values of certain parameters change rather than for the fixed values of parameters. For a virtual laboratory, this kind of mathematical statement of a problem allows modifying the conditions in which virtual experiments are conducted without changing the model of the experimental setup. If the model is developed in a standard way, on a finite set of parameter changes, a lot more calculations are required. Anecdotal evidence suggests [37–49] that applying neural networks appears to be more promising for such problems.

The gist of this approach can be explained by the simplest (in the general case non-linear) boundary problem (1). The specificity of this problem is that its statement includes the parameter $\mathbf{r} = (r_1, \ldots, r_k)$, which change at some intervals $r_i \in (r_i^-; r_i^+)$, $i = 1, \ldots, k$:

$$A(u, \mathbf{r}) = g(\mathbf{r}), u = u(\mathbf{x}, \mathbf{r}), \mathbf{x} \in \Omega(\mathbf{r}) \subset R^p, \ B(u, \mathbf{r})\big|_{\Gamma(\mathbf{r})} = f(\mathbf{r}), \tag{1}$$

where $A$ is a differential operator; $B$ is the operator that allows establishing boundary conditions; $\Gamma$ is the boundary of domain $\Omega$.

The approximate solution to the problem (1) is presented as the output of an artificial neural network with a specified architecture:

$$u(\mathbf{x}, \mathbf{r}) = \sum_{i=1}^{N} c_i v_i(\mathbf{x}, \mathbf{r}, \mathbf{a}_i), \tag{2}$$

The weights of the parameters on which the output of the neural network depends linearly and parameters $\mathbf{a}_i$ on which the output of the neural network depends non-linearly—these are determined in the course of stepwise network learning, on the basis of minimizing the error functional:

$$\sum_{j=1}^{M} (A(u(\mathbf{x}_j, \mathbf{r}_j)) - g(\mathbf{x}_j, \mathbf{r}_j))^2 + \delta \sum_{j=1}^{M'} (B(u(\mathbf{x}'_j, \mathbf{r}'_j)) - f(\mathbf{x}'_j, \mathbf{r}'_j))^2 \tag{3}$$

In the specialized books and articles on neural networks, the above mentioned process is referred to as learning.

Here $\left\{\mathbf{x}_j, \mathbf{r}_j\right\}_{j=1}^{M}$ are sampling points in the area $\Omega(\mathbf{r}_j) \times \prod_{i=1}^{k} (r_i^-; r_i^+)$; $\left\{\mathbf{x}'_j, \mathbf{r}'_j\right\}_{j=1}^{M'}$ are the sampling points on its boundary $\Gamma(\mathbf{r}'_j)$, $\delta$ is a positive parameter.

Coordinates and time (if the process is dynamic) as well as the parameters of the object (or the process), which can vary, are fed to the neural network input. The relevant physical values are fed to the output. Neural network learning involves adjusting its weights by minimizing error functional (3) by using a method of non-linear optimization. We usually use such well-known methods as RProp or cloud + RProp in our research. Test points are regenerated after every 3–5 steps of non-linear optimization.

The numerical experiments that we have conducted illustrate that using a fixed set of test (sampling) points $\left\{\mathbf{x}_j, \mathbf{r}_j\right\}_{j=1}^{M}$ does not make sense because in this case the error functional can be small, but it can entail major errors in other points (different from the sampling ones) in area $\Omega$.

We have suggested the solution to this problem which involves using the sets of sampling points [21–27] which are periodically generated again. The repetitive generation of the sampling points after a few steps of the network learning process makes this process more sustained.

This approach has been tested on numerous problems that involve simulation of the real objects, and it has proved to be quite universal.

We have already seen that creating virtual laboratories as well as teaching in them involves drawing on a lot of expertise in different spheres: mathematics, physics and professionally-oriented subjects. The experimental laboratory course should perfectly dovetail with the rest of the curriculum, CLIL approach being an example of such interdisciplinary integration.

### 1.1.4. CLIL Approach and Its Applications in Tertiary Education

CLIL (content and language integrated learning approach) has been adopted by many educational institutions over the past decade [50–63]. It has become so topical because a lot of attention is being given to interdisciplinary interaction and knowledge integration in general. CLIL is used at schools, vocational colleges as well as in tertiary education when a number of professional subjects are explained in a foreign language, priority being given to job-relevant learning [54–68]. In our research, we mainly concentrate on adopting CLIL approach in higher education.

Another reason why CLIL is gaining popularity is that is apparently motivating for learners [44,45]. In a CLIL class, instead of learning the subject matter and acquiring foreign skills in isolation, students become more engaged in the cognitive activities by participating in interdisciplinary experiences therefore they see the value of what they are learning [60]. Thus, adopting the CLIL approach will strengthen the motivation of those engineering students who are unwilling to focus on language learning. However, traditionally, a foreign language is a separate subject on the curriculum, and it is often referred to as "a foreign language for special purposes". In the past, students had to translate a considerable number of articles on the subject they specialized in [61,62].

Then in the 1980s course books for teaching foreign languages to engineering students specializing in electronics, IT technologies, etc. were published. At the time, such course books were seen as a paradigm shift in teaching a foreign language for special purposes because, apart from reading professional texts, they had listening comprehension, vocabulary, discussion sections as well as ample opportunities to practice writing. The outcome of this approach to foreign language teaching was far more positive because engineering students did not just learn to read a foreign-language article on their subject and recognize the familiar vocabulary, but also to have professional conversations with their foreign colleagues. Nevertheless, sometimes university teachers still had difficulty motivating their students to learn a foreign language because they recognized it as an unimportant subject.

CLIL was the next crucial step, which helped change engineering students' attitude to foreign language and increased their motivation. In teaching a foreign language for special purposes, the focus is on language learning, whereas in CLIL more attention is given to the job-related subjects. Therefore, when the students are aware of the fact that they study the subjects that are essential for their future career, they are more willing to get involved in cognitive activities than when they study a subject that they perceive as insignificant.

Another considerable advantage of CLIL is that students do not focus on learning professional vocabulary, which is seen by most students as an insurmountable obstacle. Instead, they concentrate on doing their assignments, such as solving equations, conducting experiments and doing calculations. However, in CLIL class, everything is taught in a foreign language: the students ask and answer questions, the teacher gives explanations and instructions, etc. Therefore, students are able to overcome their main stumbling block in language learning: they start using the foreign-language professional vocabulary in the course of communication. Thus, the new professional terms as well as general academic vocabulary (especially if students are frequently exposed to these words) are internalized subconsciously.

In spite of all the advantages mentioned above, CLIL approach has been criticized by some scholars and educationalists. The argument goes, students with a low level of language proficiency will not be able to understand the special subject. On the other hand, the advocates of CLIL approach argue that "learners' exposure to discourse in the foreign language should be gradual, starting at a more passive, receptive level, just limited to listen to the teacher's instructions and simple concept checking questions, with the scaffolding of short-length texts, vocabulary lists, information organizers and visual aid" [63]. It means that even the students with rudimentary knowledge of a foreign language will be able to understand the instructional material. In this way, the students are supposed to skip the unnecessary stage of translating the content from their native tongue into a foreign language.

The following query is also frequently raised: who is supposed to teach if CLIL approach is adopted? Universities are unlikely to recruit enough teachers who are able to teach whether a language teacher has enough competence to teach professional content alongside with proficiency in a foreign language. Thus, if a non-bilingual teacher specializing in a particular professional subject gives instructions and explanations in class, language errors are inevitable. Such mistakes can affect the clarity of explanations and, consequently, the students' perception of the professional content. Moreover, they remember the teacher's mistakes on a subconscious level, which can have negative effects on their language proficiency.

In some studies, the effectiveness of CLIL is questioned because this approach does not always allow a teacher to achieve a desirable outcome. For instance, in one study CLIL was used in a history class at school. Even though CLIL had a positive effect on the receptive listening skill and an insignificant effect on productive English skills, learners needed more input to gain the same knowledge of the content than they would acquire in a conventional non-CLIL classroom [64]. Thus, some educationalists doubt whether CLIL makes sense at all since more time is needed to teach the same content than in a traditional class. One more issue to address is finding teachers who are equally knowledgeable about the subject matter and the language.

In most cases, a language teacher is not knowledgeable enough about professional content, so serious problems are inevitable in a CLIL class. For example, if students have some queries concerning the content, the language teacher might not be in a position to deal with them. The teacher can even make language errors because he/she is not necessarily familiar with the professional terms. In many cases, students specializing in a certain subject are even more knowledgeable about terms than the language teacher, especially when dealing with the words that do not frequently occur in general English. Such deficiencies can result in awkward situations in class, and mistakes made by the teacher can shake students' confidence. A CLIL teacher is supposed to have interdisciplinary knowledge, so educational institutions interested in adopting should set up provisions for short, medium, and long-term teacher training [65]. The extensive knowledge of the subject has always been a very important gauge for the quality of teaching. Thus, if the students notice the teacher's mistakes in class and feel that he/she lacks knowledge of the subject matter, it can tarnish his/her image and result in a lack of students' motivation. Therefore, teacher training should be given a lot of attention.

One of the solutions to the problem of training the most suitable teacher capable of using CLIL in class is organizing "regular meetings between content area teachers and language specialists. In those meetings, content teachers were given some guidelines on how to structure and plan a didactic unit within the CLIL approach" [66]. The same tandem can result in knowledge exchange and cross-fertilization between language specialists and professional content providers.

Practical evidence suggests that, in general, interdisciplinary tandems at any level within universities prove to be very useful. For example, at Peter the Great St. Petersburg Polytechnic University post-graduate engineering students translated articles of a highly specialized nature in tandem with linguistic students. This cooperation appeared to be effective in producing a high-quality bibliographical review, and, according to the opinion survey, all the students who worked in collaboration were satisfied with it because they benefitted from it. Working in tandem allowed

the students specializing in linguistics to acquire technical terms and learn the basics of particular engineering subjects, whereas engineering students improved their foreign language proficiency.

Another example of interdisciplinary knowledge exchange can be observed at some universities when a group of scientists from different departments join their efforts to write an article in an academic journal. Linguists prepare a bibliographical review and work on the translation of the article into English whereas the scientists from engineering departments come up with the professional content. Similar knowledge exchange can take place if university teachers from engineering and foreign language departments collaboratively prepare for classes in the CLIL format. Engineering specialists can explain to linguists which technical terms to use and share their knowledge of the content with them. Linguists, in their turn, can teach engineering specialists how to express their ideas in English and correct any mistakes the latter make. However, this kind of collaboration between linguists and engineering specialists may not be sufficient for either of these groups to adopt the CLIL format because teaching at university requires considerable insights about the subject, which can only be provided by formal education.

In some situations, adopting the CLIL approach can yield impressive results. For instance, scientists from other countries communicating with the students in a foreign language can teach the subject they specialize in, or a bilingual local teacher who did a course in a particular subject is capable of teaching it in the CLIL format. However, such situations are quite rare and universities do not have teachers who have extensive content knowledge alongside with considerable language skills. Some educationalists have managed to strike the optimum balance, which seems to be one of the possible solutions to the problem. They claim that CLIL is only recommendable if a language teacher has enough knowledge of the basics to teach the induction course in a particular field [66]. In order to become familiar with the induction course, language teachers can do a further training course in a particular engineering subject or receive counseling and advice from content teachers. Students can also benefit from doing the induction course in a foreign language because they will learn definitions of the basic terms and it will be easier for them to succeed in vocabulary acquisition.

Another solution could be adopting CLIL approach by conducting classes in a virtual laboratory, which can be defined as the use of computer's hardware and software to generate a simulation of an experiment [67]. Not only does a virtual laboratory provide visibility and interactivity, but it also creates the environment for active learning. Instead of passively receiving information, students participate in cognitive activities. A virtual laboratory provides learners with a toolkit of virtual components and tools that are necessary for doing the assignment, for instance, conducting an experiment. In our article, we suggest that English should be used as the working language in the course of doing the assignment.

Theoretical pedagogical background of virtual laboratories is connected with the concept of virtual language learning environment [68]. According to this concept virtual worlds are playing an increasingly important role in education, especially in language learning. In 1999, Philip Rosedale formed Linden Lab with the intention of developing computer hardware to allow people to become immersed in a virtual world. Joe Miller, Linden Lab Vice President of Second Life Platform and Technology Development, in 2009 claimed that "language learning is the most common education-based activity in Second Life" [69].

The following linguodidactic advantages of virtual worlds can be defined as the most important ones: immersiveness, collaboration, task-based learning, and game-like opportunities:

- Immersiveness: immersive experiences draw on the ability to be surrounded by a certain (real or fictitious) environment that can stimulate language learning.
- Collaboration: almost all 3D virtual spaces are inherently social environments where language learners can meet others, either to informally practice a language or to participate in more formal classes.
- Task-based learning and creativity: an approach to language learning in virtual reality which involves construction of objects as part of a language learning activity [70].

It should be emphasized that the idea of immersive opportunities of the virtual worlds is becoming more and more important nowadays given the frenetic pace of development of online education, which was dictated by the pandemic situation in the whole world.

Having analyzed different virtual world approaches and educational tools which are typical for creating language learning environments we can say that the blended learning approach is the best one for this purpose. The language learners here are exposed to a 3D virtual environment for a specific activity or time period. This approach allows combining the use of virtual worlds with other online and offline tools. The following tools can be used for this purpose:

(1)　MOODLE (Modular Object-Oriented Dynamic Learning Environment)—free and open-source learning management system, based on pedagogical principles.
(2)　SLOODLE (Simulation Linked Object Oriented Dynamic Learning Environment)—an open-source project which integrates the multi-user virtual environments of Second Life and OpenSim with the Moodle learning-management system.
(3)　Languagelab.com and avatar Languages—a complete language learning environment through a virtual world.

In general, we are supposed to make a conclusion that virtual world language learning can offer distinct (although combinable) learning experiences.

Coming back to the system of higher education and speaking about content and language integrated learning at universities, we should state that theoretical and practical implementation of the ideas of virtual content and language-learning environment is only in its infancy. Independent investigations are being carried out. For example, the aspects of using MOODLE or MOOCs (massive open online courses) are deeply studied and analyzed in pedagogics and language methodology [71–74].

But there are no complex studies of the problem which is stated in the title of our article: how to develop a model of increasing the learners' bilingual professional capacity in the virtual laboratory environment.

The main advantage of using CLIL in a virtual laboratory is that in the course of doing a practical assignment students get visual aids, which help them retain information. Instead of trying to memorize lists of foreign terms, learners hear the professional vocabulary from the teacher and use them in the course of reading the instructions related to the laboratory assignment. In addition, since they see the whole process going on as if it was in the real world, they link the new professional terms directly with the images corresponding to them. Thus, they skip the unnecessary stage of translating the terms from English into their native language.

Working in a virtual laboratory can be regarded as a type of scaffolding, which is the key condition for using CLIL. Two types of instructional scaffolding—contextualizing and schema building—are relevant in a virtual CLIL class. Contextualizing can be defined as adding context to academic language (for example, films and images). Schema building involves providing thinking frameworks (such as charts and organizers) to help illustrate ideas. In a virtual laboratory, the context is added by providing virtual tools and components in the form of computer screen images. Working with them, the learners do the laboratory assignment and write a performance report in English. They sometimes have to describe all the steps of a laboratory assignment by attaching graphs to their report. In this way, students do not only subconsciously learn the terms in the course of doing the laboratory assignment, but also revise them while they are writing a performance report. In other words, virtual laboratories present a new type of technological means (virtual technologies) possessing huge amount of didactic possibilities for content and language acquisition.

In order to develop the approaches on which this laboratory can function we decided to join together some of the above mentioned ideas of CLIL and virtual technologies existing in the Russian higher education institutions and tried to present them as blocks of the model which increase the learners' bilingual professional capacity in the virtual laboratory environment. The model can be classified in the following way: (1) The laboratory block for supporting professional-oriented training

in a foreign language. (2) The laboratory block for supporting language training connected with professional content. (3) The laboratory block for bilingual and trans-professional training on the basis of professional metalanguage.

In other words, our purpose is to propose new tools for content and language integrated learning which would be incorporated into the blocks of the model.

1.1.5. The Model of a Virtual Bilingual Professional Laboratory

The laboratory block for supporting professional-oriented training in a foreign language.

Training in a foreign language in this case is carried out on the basis of subject-oriented training materials; actually, language textbooks turn into the category of reference books. Acquisition of a foreign language within the model in question is carried out at a higher level of students' motivation and interest. Foreign language mastery is achieved through conversational practice, which students get in the course of understanding and discussing the professional-oriented subject matter in a foreign language [74,75].

In the first case, the following purpose of teaching is defined: specification of the teaching content, development of tutorials, improving the quality of teaching and students' learning efficiency of professional focused language. Adopting this approach requires development of joint educational and methodical complexes on the professional language: textbooks, glossary, audiovisual materials, laboratory works, a set of control and measuring materials.

The laboratory block for supporting language training connected with professional content.

In this case, bilingual training in a particular subject matter is provided by means of foreign and native languages: the bilingual subject competence is gained by studying the subject matter. The following teaching outcome is achieved: the foreign language is used along with the native language as the means of cognitive activity aimed at acquiring the subject-oriented (professional) knowledge in the course of professional training of future specialists. The principles of this approach are the following: (1) the principle of using both the native language and the foreign one in the course of gaining the subject knowledge; and (2) the principle of unity of cognitive and speech activities in a foreign language.

The laboratory block for bilingual and trans-professional training on the basis of the professional metalanguage.

According to our approach, vehicle languages of individuals or groups work together as one socio-cognitive-linguistic unit for purposes of communication and education. Accordingly, any specific constellation of natural and technological languages should be treated and researched as a unit.

It has long been recognized that new languages are most easily acquired not before but while being engaged with professional challenges—in the course of social interactions in the teaching and learning process. Metalanguage in this approach goes further by recruiting for content and language(s) integrated learning the linguistic structures or compositional grammar including texts and discourses that are inherent in professional (in our case engineering) education.

At the next level of our research the experiment was conducted in 2019 at the Higher School of Media-Communication and PR, Peter the Great St. Petersburg Polytechnic University.

A special virtual laboratory was created for the course of International Business that is taught in English. Students work at the laboratory during the first stage of integrated learning [76–80]. The laboratory contains video materials, case studies and exercises for the initial training and lecture preparation. At the third stage of integrated learning, after the face-to-face lecture, students are involved in online international project X-Culture and work on real projects in the virtual intercultural teams. In the course of carrying out the project, the students communicate in English with their team members from foreign universities using various online channels—skype, e-mail, WhatsApp, etc.

## 2. Materials and Methods

To evaluate the efficiency of virtual laboratory environment partly based on the digital twin technology, we analyzed the students' proficiency in English as well as their professional attainments in the subject. The obtained data allowed us to reveal whether the virtual laboratory environment influenced students' outcomes in CLIL group. For the analysis we compare the results of two groups of students—Group A, students studied on the basis of integrated learning with elements of virtual environment (N = 23) and Group B (N = 25) (students studied professional discipline and professional English separately).

Both groups were tested before the experiment to identify the level of English proficiency. The test consisted of four parts: listening, reading, writing, and speaking. Partly the test was conducted through the online platform Moodle that was developed for the St. Petersburg Polytechnic University (listening, reading, writing) and partly at seminars (speaking). In Part 1, students listened to the audio recording twice, and then gave short answers to the questions related to it. In the reading section, the students were offered a text with a volume of 500 words. After reading the text, it was necessary to complete three tasks: complete the text with correct topic sentences for each paragraph, choose whether the statements are true/false and answer the questions about the gist of the text. In the writing section, a cohort of the students wrote a formal business letter. For the speaking test, students were arranged in pairs. This made it possible to assess the peer interaction and to reduce the testing time. Concluding the experiment, at the end of the semester, students again underwent assessment testing.

To assess the knowledge of the professional subject (International Business), students of both groups were given a final test consisting of 25 closed questions. All the students passed these tests through the online platform Moodle, which was developed for the St. Petersburg Polytechnic University. The students from Group A did the test in English, while the students from Group B did it in the native language (Russian).

As we wanted to implement a comprehensive assessment of the virtual environment, all the students participating in the experiment were offered to complete a motivation questionnaire anonymously. All the students passed these tests through the online platform Moodle. The closed-ended questions were designed to explore the students' perceptions about teaching and learning, defining five indicators: desire for learning English after university (Q1–Q2), anxiety (Q3–Q4), positive attitude to English (Q5–Q6), self-esteem (Q7–Q8) and self-demand (Q9–Q10).

For the analysis descriptive statistics, and pair-samples Students' *t*-test were conducted.

## 3. Results

### 3.1. English Testing

Figures 1–4 shows the results of the initial testing and final testing of the two groups for comparison of the results obtained. The pre-test results of the two groups are approximately the same (the Group A pre-test and Group B pre-test lines practically merge). The results of the post-tests for Group A are higher in all categories (reading, writing, listening and speaking).

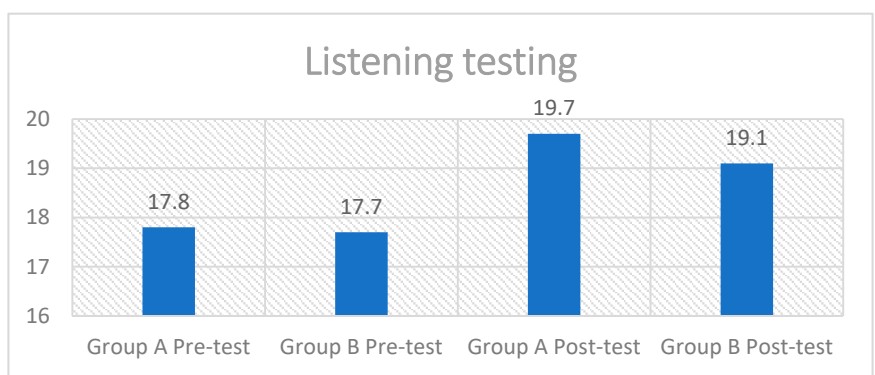

**Figure 1.** Listening testing results.

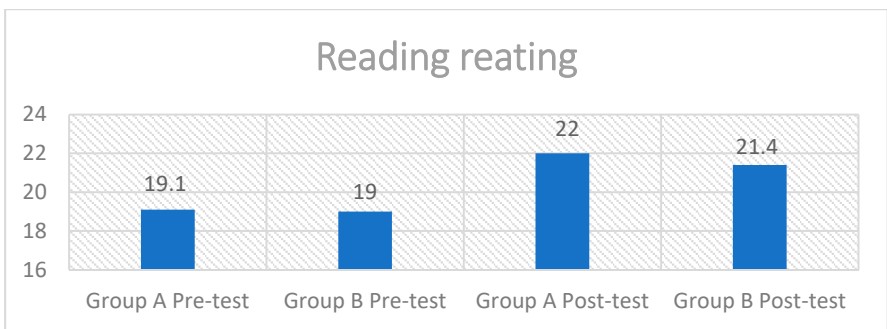

**Figure 2.** Reading testing results.

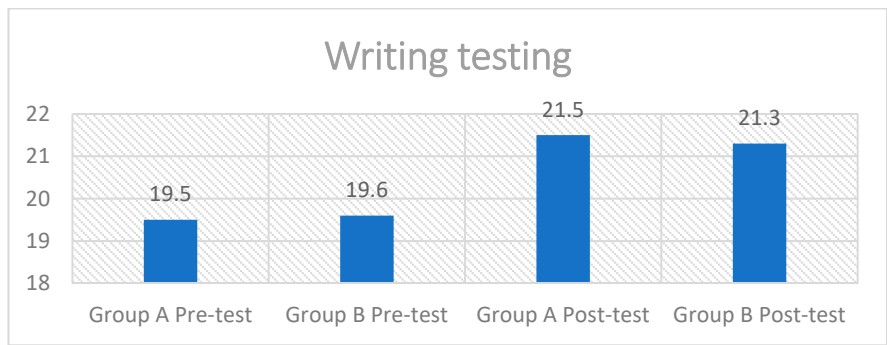

**Figure 3.** Writing testing results.

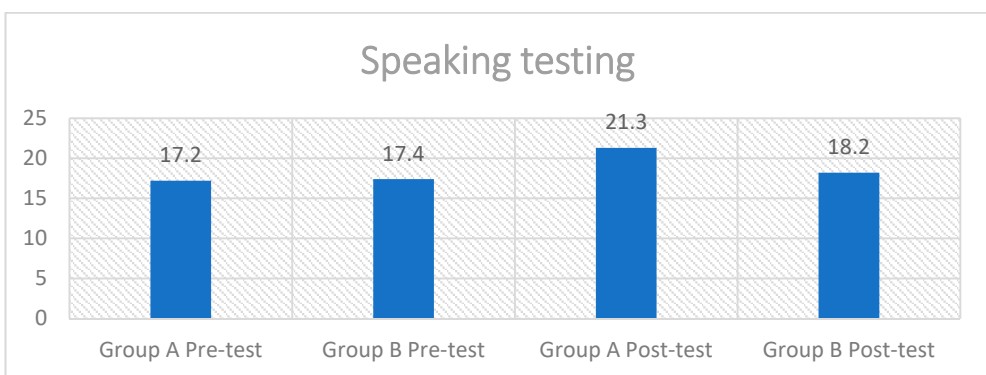

**Figure 4.** Speaking testing results.

In general, the overall results of students' English proficiency have improved in both groups. However, comparison of the results of the two tests (before and after the course) done by all the

experiment participants indicates that improvements in Group A (the students who studied on the basis of integrated approach) were more significant (Table 1). Hence, we can confirm the efficiency of integrated learning in the virtual environment for English learning purposes.

**Table 1.** Descriptive statistics (results of pre-test and post-test on English).

| Category | Group | Test | Results (Average Mean) | SD | *t*-Value |
|---|---|---|---|---|---|
| Listening | Group A | Pre-test | 17.8 | 2.16 | 3.1 ** |
| | | Post-test | 19.7 | 2.3 | |
| | Group B | Pre-test | 17.7 | 2.43 | 2.5 * |
| | | Post-test | 19.1 | 1.98 | |
| Reading | Group A | Pre-test | 19.1 | 2.18 | 5.4 *** |
| | | Post-test | 22 | 2.12 | |
| | Group B | Pre-test | 19 | 1.99 | 4.6 *** |
| | | Post-test | 21.4 | 2.17 | |
| Writing | Group A | Pre-test | 19.5 | 2.01 | 3.0 ** |
| | | Post-test | 21.5 | 1.98 | |
| | Group B | Pre-test | 19.6 | 2.21 | 2.7 * |
| | | Post-test | 21.3 | 2.12 | |
| Speaking | Group A | Pre-test | 17.2 | 2.42 | 5.8 *** |
| | | Post-test | 21.3 | 2.34 | |
| | Group B | Pre-test | 17.4 | 2.55 | 1.2 |
| | | Post-test | 18.2 | 2.15 | |

Note: * $p < 0.05$; ** $p < 0.01$; *** $p < 0.001$.

### 3.2. Professional Discipline Testing

International business testing results of both groups are presented below (Table 2). In addition, we calculated the *t*-value to determine whether the difference between the results of the two groups was significant.

**Table 2.** Descriptive statistics (professional discipline testing).

| Test | Group | Results (Average Mean) | SD | *t*-Value |
|---|---|---|---|---|
| Final testing on professional discipline | Group A | 69.81 | 6.5 | 1.6 |
| | Group B | 70.9 | 7.12 | |

Even though the results of Group A are slightly lower than the results of Group B, the difference between the results is not significant. Therefore, we can conclude that an integrated approach is efficient not only for English learning purposes but for professional discipline learning as well.

### 3.3. Motivation Testing

Both groups of students were asked to pass the online survey on motivation at the end of the semester. The results of the survey measured by 10 point Likert scale are presented in Figure 5.

The students who did the integrated learning course with the elements of a virtual laboratory were more motivated for English learning after university and have a positive attitude to English. Moreover, their level of anxiety was significantly lower, which proves the positive influence of the integrated approach and the virtual laboratory.

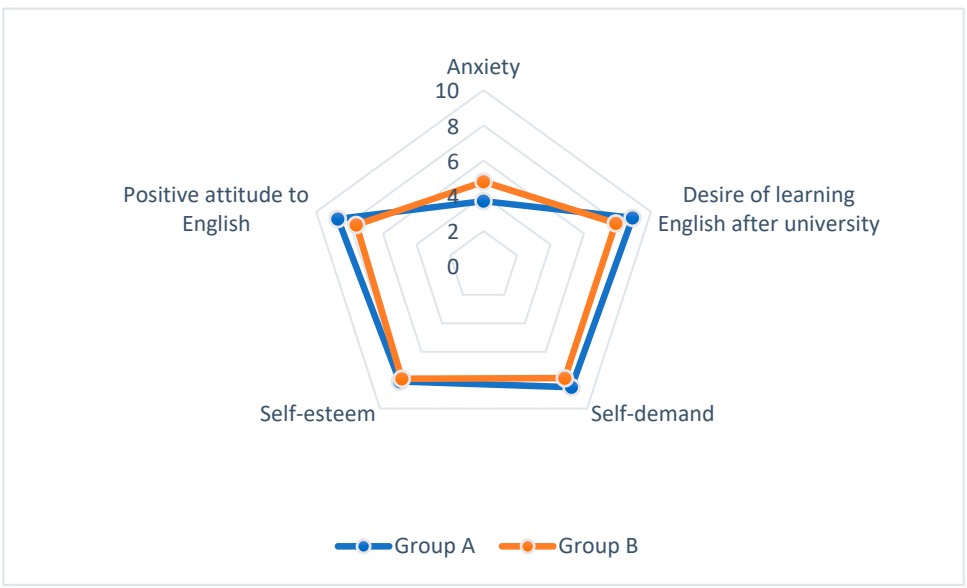

**Figure 5.** Results of students' motivation survey.

## 4. Discussion

Virtual laboratories should be widely used not only in industry, but also for educational purposes because they prompt learners to revise their experimentation strategies and to reflect on their findings in science courses. A neural-network-based approach is worth using for the simulation of physical processes in a virtual laboratory since neural networks are not affected by data errors. Besides, natural parallelization of computations is possible.

Previous studies of CLIL implementation in the educational process [61,64,65] showed the impressive results of learning English and developing students' communication skills, while some studies reflected various limitations of using CLIL approach and indicated poor professional discipline mastering. We, in turn, offer a new educational model based on virtual laboratory that can supplement and enhance CLIL learning.

Conducting experiments in virtual laboratories help engineering students gain a more nuanced realizing of scientific phenomena and more thorough understanding of inquiry. The research has shown that combining CLIL (integrated learning) strategies with working in a virtual laboratory strengthens students' motivation and stimulates their interest in learning English because this approach offers students such advantages as immersiveness, collaboration, task-based learning and game-like opportunities.

*Limitations of the Present Study and Suggestions for Future Research*

The theoretical work of other scholars in this field has been a useful resource for planning and designing, and we expect that our study will provide something of value for future researchers, too. Of course, there are some limitations in our study, as it does not take into account a novelty effect—students did not have an experience of learning in a blended environment. Additionally, the sample size was relatively small because it was the first time, we implemented such an educational model, and the duration of the course was only one semester.

In our further research, we are going to evaluate students' satisfaction of virtual laboratory in an integrated learning model and upgrade it due to responses.

This study was conducted within the federal project "The Development of Scientific and Research and Production Cooperation" under the national project "Science". It is aimed at setting up a scientific-technological center "Mathematical Methods and Intelligent Control System" in the direction

"Perspective Platform Solutions for the Integration of Industrial Technologies of Cyber-Physical Systems and Artificial Intelligence Systems".

**Author Contributions:** Conceptualization: L.P.K. and D.A.T.; data curation: A.M.K.; formal analysis: A.M.K. and A.I.D.; investigation: A.M.K. and M.A.O.; methodology: L.P.K.; project administration: L.P.K.; resources: A.I.D.; supervision: M.A.O.; validation: A.M.K. and M.A.O.; writing—original draft: A.M.K. and A.I.D.; writing—review and editing: D.A.T. and L.P.K. All authors have read and agreed to the published version of the manuscript.

**Funding:** This research work was supported by the Academic Excellence Project 5-100 proposed by Peter the Great St. Petersburg Polytechnic University.

**Conflicts of Interest:** The authors declare no conflict of interest.

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
