# Peer review of "Developing a Model of Increasing the Learners’ Bilingual Professional Capacity in the Virtual Laboratory Environment"

_applsci, doi:10.3390/app10207022_

Round 1
Reviewer 1 Report
The authors have written an extremely interesting article combining various approaches while integrating the CLIL approach at university level. This research should be published because of its originality and its intricate approach to teaching high level technology as well as foreign languages. Moreover, they make a valuable contribution for introducing CLIL at university level. For all this, I would like to highly praise them.
Author Response
Thank you very much!

Reviewer 2 Report
The research is interesting and important. However, I have some remarks to its description:
- The intro and methodological/experimental parts are very imbalanced. The intro is very interesting (full of catchy terminology as digital twins, neural networks) and, therefore, raises a lot of expectations for the methodological/experimental part.
- There is no clear explanation how do you use a neural network: what is on its input and output, how do you train it?
- Lines 557-558 “The results of the post-tests for group B are higher in all categories (Reading, Writing, Listening and Speaking)”. For me it seems opposite, if not, please, explain the meaning of y axis in Fig. 1-Fig. 4.
- In line 567 you say that “In general, the overall results of students’ English proficiency have improved in both groups”; besides, in lines 544-545 you state that “The students from group A did the test in English, while the students from group B did it in the native language (Russian)”. If students in group B did the test in Russian, how can you measure that their English proficiency improved?
Author Response
The intro and methodological/experimental parts are very imbalanced. The intro is very interesting (full of catchy terminology as digital twins, neural networks) and, therefore, raises a lot of expectations for the methodological/experimental part.
Thank you for your comment. The introduction of a virtual laboratory based on digital twin technology into the educational process is a very hard and long process. We offered a mathematical solution that was partly tested in the framework of the CLIL course. Before the lectures students were preparing at the university virtual laboratory, the program was set up so that when choosing the wrong answers, the student received similar tasks. The goal was to achieve a high level of assimilation of the material. So, to evaluate the efficiency of using such a virtual laboratory we conducted the study of students’ educational performance on English and professional discipline.
There is no clear explanation how do you use a neural network: what is on its input and output, how do you train it?
Thank you for your comment. Coordinates and time (if the process is dynamic) as well as the parameters of the object (or the process), which can vary, are fed to the neural network input. The relevant physical values are fed to the output. Neural network learning involves adjusting its weights by minimizing error functional by using a method of non-linear optimization. We usually use such well-known methods as RProp or cloud+ RProp in our research. Test points are regenerated after every 3-5 steps of non-linear optimization.
Lines 557-558 “The results of the post-tests for group B are higher in all categories (Reading, Writing, Listening and Speaking)”. For me it seems opposite, if not, please, explain the meaning of y axis in Fig. 1-Fig. 4.
Thank you for your comment. It was a mistake in the text. Group A (studied an integrated learning course had higher results at the end of the semester.)
In line 567 you say that “In general, the overall results of students’ English proficiency have improved in both groups”; besides, in lines 544-545 you state that “The students from group A did the test in English, while the students from group B did it in the native language (Russian)”. If students in group B did the test in Russian, how can you measure that their English proficiency improved?
Thank you for your comment. In the lines 540-544 we stated that the test on professional discipline (International business) was on different languages for groups A and B (as they studied the course based on different educational approaches). But the English testing was on English. It was two different tests.
